# A Robust Model for Circadian Redox Oscillations

**DOI:** 10.3390/ijms20092368

**Published:** 2019-05-13

**Authors:** Marta del Olmo, Achim Kramer, Hanspeter Herzel

**Affiliations:** 1Institute for Theoretical Biology, Charité and Humboldt-Universität zu Berlin, 10115 Berlin, Germany; 2Laboratory of Chronobiology, Charité-Universitätsmedizin Berlin, 10117 Berlin, Germany; achim.kramer@charite.de

**Keywords:** redox, oscillations, mathematical modeling, negative feedback, fast vs. slow reactions, phases, switches

## Abstract

The circadian clock is an endogenous oscillator that controls daily rhythms in metabolism, physiology, and behavior. Although the timekeeping components differ among species, a common design principle is a transcription-translation negative feedback loop. However, it is becoming clear that other mechanisms can contribute to the generation of 24 h rhythms. Peroxiredoxins (Prxs) exhibit 24 h rhythms in their redox state in all kingdoms of life. In mammalian adrenal gland, heart and brown adipose tissue, such rhythms are generated as a result of an inactivating hyperoxidation reaction that is reduced by coordinated import of sulfiredoxin (Srx) into the mitochondria. However, a quantitative description of the Prx/Srx oscillating system is still missing. We investigate the basic principles that generate mitochondrial Prx/Srx rhythms using computational modeling. We observe that the previously described delay in mitochondrial Srx import, in combination with an appropriate separation of fast and slow reactions, is sufficient to generate robust self-sustained relaxation-like oscillations. We find that our conceptual model can be regarded as a series of three consecutive phases and two temporal switches, highlighting the importance of delayed negative feedback and switches in the generation of oscillations.

## 1. Introduction

The Earth’s regular 24 h rotation has led to the evolution of circadian oscillators in all kingdoms of life. These oscillations control daily rhythms in metabolism, physiology, and behavior, and they allow organisms to adapt their physiological needs to the time of day in an anticipatory fashion [1,2]. When the internal clocks run in synchrony with the external world, they provide organisms with significant competitive advantages [3].

Although the molecular clockwork components have widely divergent origins and are not conserved across the main divisions of life, a common design principle has been applied to all organisms where circadian timing mechanisms have been investigated. This paradigm relies on a negative transcription-translation feedback loop (TTFL), where protein products of clock genes feed back periodically to regulate their own expression and drive rhythmic output pathways and physiology [2,4,5]. Mounting evidence suggests that transcription-based oscillators are not the only means by which cells track time. Some examples of non-transcriptional oscillators are (i) the cyanobacterial phosphorylation oscillator, which can be reconstituted in vitro [6]; (ii) the circadian photosynthetic rhythms that persist in green algae after enucleation and hence in the absence of nuclear transcription [7]; or (iii) the peroxiredoxin oxidation rhythms found in red blood cells [8], which are anucleate.

Peroxiredoxins (Prx) are a conserved family of antioxidant enzymes that maintain the cellular redox state by clearing the cell from reactive oxygen species (ROS). They reduce hydrogen peroxide (H_2_O_2_) to water with the use of reducing equivalents provided by other physiological thiols [9]. As a result ROS removal, Prxs become oxidized in their active site [9]. Interestingly, and in contrast to the divergent evolution of the TTFL, levels of oxidized Prx have been shown to oscillate with a circadian period in all kingdoms of life [10]. The exact mechanism that generates circadian Prx redox rhythms, however, still remains an open question, and it seems to be different in different cell types (Figure 1A). Sulfiredoxin (Srx), the enzyme that reduces oxidized Prx, is likely a key determinant of Prx3 hyperoxidation rhythms in mammalian adrenal gland, heart, and brown adipose tissue [11]. In red blood cells from Srx knockout mice, however, Prx redox oscillations persist, and rhythms have been shown to be dependent on the degradation of the oxidized enzyme by the proteasome [12]. Moreover, the presence of Prx oscillations in organisms that lack Srx homologs such as *C. elegans* or *N. crassa* indicates that Prx reduction is, at least in some cases, dispensable for the cycle to occur.

In this study, we use mathematical modeling to investigate the principles for Prx/Srx oscillations. Our results show that the combination of a fast Prx hyperoxidation event followed by a slow and delayed negative feedback loop is the minimal backbone that is required for the system to oscillate. We also find that this minimal motif produces relaxation-like oscillations with two temporal switch-like events, highlighting the importance of switches in the generation of oscillations.

## 2. Results

### 2.1. Derivation of a Kinetic Model for Circadian Redox Oscillations

Because of their potential to oxidize and damage cellular protein and lipids, ROS levels must be under tight regulation. Prx3 is one of the major antioxidant proteins involved in H_2_O_2_ removal in mitochondria. A conserved cysteine (Cys) residue in its active site is oxidized by H_2_O_2_ to Cys-sulfenic acid (Prx3-SOH, reaction $k_{1}$ in Figure 1B). This intermediate can react with a second conserved Cys from another Prx3 subunit to produce an intermolecular disulfide bond (reaction $k_{2}$), which can be reduced by the thioredoxin (Trx) system of the cell (reaction $k_{3}$). Alternatively, Prx3-SOH can undergo further oxidation, termed hyperoxidation, in an S-sulfinylation reaction, to form a sulfinic acid (Prx3-SO_2_H, reaction $k_{4}$) [9,13]. Prx3-SO_2_H is catalytically inactive [9,13], and in adrenal gland cells, brown adipocytes or cardiomyocytes, it can be reduced back to Prx3-SOH by action of sulfiredoxin (Srx) (reaction $k_{5}$) [11]. In other cell types and other model organisms, the Srx-mediated reduction of Prx-SO_2_H is not required for rhythm generation [12].

In mitochondria from brown adipose, adrenal gland and heart tissue, Prx3-SO_2_H and Srx levels oscillate with a circadian period [11]. H_2_O_2_ increase results in the oxidation and inactivation of Prx3 (reactions $k_{1}$ and $k_{4}$ from Figure 1B). The consequence is the accumulation of H_2_O_2_ inside mitochondria and its overflow to the cytosol ($k_{6}$ in Figure 1B). Cytosolic H_2_O_2_ activates pathways to control its own production. Among others, it stimulates Srx oxidation and intermolecular disulfide bridge formation with Hsp90 (reaction $k_{7}$) to promote translocation of Srx to the mitochondria (reaction $k_{8}$). Mitochondrial Srx can reduce and thus reactivate Prx3-SO_2_H (reaction $k_{5}$) [11]. Srx levels peak in mitochondria ∼8 h after Prx3-SO_2_H, and Srx becomes sensitive to degradation (reaction $k_{9}$) when Prx3-SO_2_H levels decrease [11]. The mitochondrial import of Srx thus constitutes a negative feedback that enables a new cycle of H_2_O_2_ removal and Prx3-SO_2_H accumulation (Figure 1B).

In order to understand how fast biochemical redox reactions result in slow 24 h rhythms, we develop a deterministic model containing the biochemical species from the Prx3/Srx system shown in Figure 1B. The complete set of equations of this large model, which we refer to as *detailed model*, are found in Appendix A. Unfortunately, quantitative details of the kinetic processes are not known and thus estimating parameters in such a large model represents a challenge. For this reason, and in order to gain insight into the design principles of redox rhythms in the Prx3/Srx system, we simplify the detailed model to its core motif (Figure 2A). Details of the model reduction are found in Appendix B.

### 2.2. A Low Dimensional Model Represents the Core Prx3-SO_2_H/Srx Circadian Oscillator

We identify the *core* motif that is required for generation of Prx3-SO_2_H and mitochondrial Srx circadian oscillations by systematically changing parameters and clamping variables to their mean value [14] (Appendix B). The scheme and equations of this minimal model are shown in Figure 2. It only contains the variables, kinetic parameters, and reactions that are necessary and sufficient for self-sustained rhythm generation.

The core oscillator model consists of only five variables and just one oxidation event (Figure 2A). *A* represents the *active* but partially oxidized Prx3 (Prx3-SOH) that gets further hyperoxidized by the *danger 1* ($D_{1}$, mitochondrial H_2_O_2_) to produce the *inactive* Prx3-SO_2_H, *I*. As the A→I reaction occurs, *A* levels decrease. The result is that the ability of mitochondria to remove $D_{1}$ also decreases. When mitochondria can no longer eliminate the oxidant, $D_{1}$ accumulates and overflows to the cytosol, where it is referred to as *danger 2* ($D_{2}$, cytosolic H_2_O_2_). $D_{2}$ activates the translocation of the *rescuer R* (mitochondrial Srx), that reduces *I* back to *A*, closing the negative feedback loop. The Prx3-SH oxidation to Prx3-SOH and the $D_{2}$-induced oxidation of cytosolic Srx (reactions $k_{1}$ and $k_{7}$ in Figure 1B) were found to be dispensable for rhythm generation. Thus, these biochemical species, as well as the reactions they take part in, are omitted from the minimal model.

We formulate ordinary differential equations (ODEs) to describe the behavior of the system over time. Production and removal terms are modeled with mass action kinetics (Figure 2B). The nonlinear term in the model arises from the quasi-steady state approximation [15,16] performed on the equation for variable *A*, whose production and removal reactions are modeled as bilinear terms (details provided in Appendix B). We assume constant $D_{1}$ production (parameter *p* in the model) and constant Prx3 pool A+I over time. Moreover, we only consider the negative feedback performed by *R*-induced reactivation of *A*, which is stimulated by the $D_{2}$ increase. However, other additional feedback loops might exist, as seen by studies that have shown that cytosolic H_2_O_2_
$D_{2}$ can feed back and decrease $D_{1}$ production by activation of the stress-activated protein kinase p38 MAPK, among other pathways [17].

### 2.3. Design Principles of the Redox Oscillator: Fast *A* Inactivation Followed by a Slow Negative Feedback Loop

The dynamics of a system depend greatly on parameter values. Unfortunately, most kinetic rates in the Prx3/Srx sytem are not known in quantitative detail and therefore finding reasonable parameters represents a major challenge. Nevertheless, some physiological constraints can be taken into account to narrow down the plausible range of parameter values. First of all, the system needs to oscillate with a circadian period and the phase relationship between *I* and *R* should be approximately 8 h, according to previously published data [11]. Other constraints come from enzymatic assays performed with Prx3 and Srx, as well as from studies that have focused on H_2_O_2_ signaling and transport [11,18,19,20,21,22,23,24]. They have estimated rates of Prx3 hyperoxidation, Srx-mediated Prx3-SO_2_H reduction, and Srx degradation and have allowed the derivation of a reasonable rate for H_2_O_2_ translocation across membranes (Table 1).

In order to obtain a reasonable set of parameters that satisfies the physiological constraints, we systematically vary all model parameters and perform bifurcation and sensitivity analyses. Bifurcation diagrams show under which parameter values the system oscillates and sensitivity analyses address how the oscillation period varies over the oscillatory range as a parameter is changed.

Bifurcation diagrams are presented for a selected choice of parameters in Figure 3A–C. The bifurcation plots show that the system oscillates for values of d,e,q<1. The limit cycle emerges at d=0.02 and disappears at d=0.5 (Figure 3A, bifurcation plots for parameters *e* and *q* are shown in Appendix C). We also find that, in contrast to small d,e,q parameter values, the system requires a 1000–10,000-fold higher *a* oxidation rate to enter into the oscillatory regime. The Hopf bifurcation occurs at a=120 (Figure 3B) and once this value is reached, oscillations persist for a wide range of kinetic parameters, with little effects in oscillation amplitude or period, indicating robustness of the model. The period sensitivity analyses reveal that the oscillation period depends strongly on the translocation rates *d* and *e* and on the *R* degradation parameter *q*, being most sensitive to changes in parameter *d* (Figure 3C and Appendix D).

These findings show that a fast *A* inactivation reaction (high *a* value) in combination with a slower negative feedback loop (1000–10,000-fold lower d,e,q rates) constitute the backbone of the redox oscillator model (Figure 3D). We thus find a plausible set of model parameter values that produce robust circadian oscillations with the expected characteristics: a period of 24.2 h and a I/R phase difference of 8.7 h, as determined by maxima estimation of the *R* and *I* dynamics (Figure 4). Moreover, the parameter choice is in agreement with the high Prx3 oxidation rate measured experimentally [18] and with the ∼10,000-fold slower physiological H_2_O_2_ translocation (Table 1).

According to our results, only a small fraction of active Prx3 *A* is needed to keep $D_{1}$ levels close to 0, as seen by the higher *I* levels compared to *A* during the whole simulated time (Figure 4). The results also predict that $D_{2}$ levels peak 3.9 h after $D_{1}$ (or 4.5 h after *I*), an observation that could be relevant in the context of the canonical TTFL and will be further addressed in the Discussion section.

Interestingly, insights from the redox oscillator model apply as well to larger and more complex models. Firstly, negative feedback loops ($D_{1}-D_{2}-R$ in the minimal model) as well as nonlinear terms (given by the *A* equation, Figure 2B) are required to achieve self-sustained oscillations [25]. Secondly, overcritical delays about quarter to half of a period are necessary for oscillations to arise [25,26,27] (in the minimal model, slow d,e,q rates are required to yield the 8.7 h delay). Lastly, translocation and degradation rates have profound effects on the period [28,29,30,31]. It should be noted, nevertheless, that the choice of parameters is not tailored to specific kinetic data. Kinetic parameters might differ among tissues and might also depend on physiological conditions.

### 2.4. The I/R Redox Oscillator is Characterized by Three Phases and Two Temporal Switches

We have shown that, in order to oscillate, the mitochondrial Prx3-SO_2_H/Srx (I/R) system requires a fast $D_{1}$-induced inactivation of *A* followed by a lengthy and slower negative feedback loop $D_{1}-D_{2}-R$ that reactivates *A* (Figure 3D). As seen by the equations from Figure 2B, the dynamics of A,I, and $D_{1}$ depend on the fast *a* oxidation rate compared to the slower dynamics of $D_{2}$ or *R*. This is reflected in the relaxation- or triangular-like waveforms of A,I, and $D_{1}$ (Figure 4). Such relaxation oscillations are generally characterized by two consecutive processes that occur on different timescales. There are intervals of time, during which little happens (A=0 or I=1 in the system’s dynamics), followed by time intervals with considerable changes. The result is the triangular-like dynamics observed for *A*, *I*, and $D_{1}$ (Figure 4).

It is known that oscillations depend on negative feedbacks, but that in the absence of appropriate delays or nonlinear terms, negative feedback circuits often settle into a stable steady state termed homeostasis [25,32,33]. Several computational studies have demonstrated that sufficiently strong nonlinearities are required to generate self-sustained oscillations [31,34,35,36]. Such nonlinear terms often exhibit sigmoidal or *switch-like* characteristics. Surprisingly, no switch-like curve is evident from the only nonlinear term of the model $ab\frac{D_{1}R}{bR+aD_{1}}$ (Figure 2B and Appendix B), nor from the response of any variable to upstream components (data not shown). Nonetheless, we find that this term is at the core of the oscillations, as its linearization results in the stabilization of the steady state. The large value of *a* allows the production of self-sustained rhythms, even in the absence of an explicit kinetic switch.

In order to emphasize the importance of the nonlinear term in the context of oscillations, and given the switch-like dynamics of A,I, and $D_{1}$, we still use the term *switch*. Instead of using it to describe how the steady state of a first component responds to a second component which is upstream of the former (often done in the literature [36,37,38,39]), we focus here on temporal switches that mark transitions between phases. We see that the model dynamics can be split into three phases separated by two temporal switches (Figure 5).

The first phase is characterized by the inactivation of *A* as a result of $D_{1}$ removal. As long as there is active Prx3 (*A*) in the system, mitochondrial H_2_O_2_ ($D_{1}$) levels are kept in check and $D_{1}$ is removed. But the cost of the $D_{1}$ clearance is the inactivation of *A* to *I*, and thus *A* levels decrease as *I* increases (first box in Figure 5B). When most of the active Prx3 pool *A* has been hyperoxidized and inactivated to *I*, the first switch occurs and the system progresses to the next phase. As a consequence of the *A inactivation*, its ability to remove $D_{1}$ molecules decreases and thus $D_{1}$ accumulates and *leaks* to the cytosol. In other words, the flux of $D_{1}$ changes from inactivating *A* to leaking, what gives this first temporal switch its name: *inactivation/leakage* (I/L) switch. As a result of the I/L switch, $D_{1}$ accumulates, leaks and activates the $D_{1}-D_{2}-R$ negative feedback. Consequently, $D_{2}$ and *R* levels increase (phase 2, second box in Figure 5B). At the end of the second phase, the increase in *R* is accompanied by a decrease in $D_{1}$. Once $D_{1}$ and *R* reach critical values in the nonlinear term, the negative feedback becomes effective and the dynamics of *A*, that are governed by the nonlinearity (Figure 2B), change suddenly. This triggers the *activation* (A) switch and the sharp increase in *A* levels (phase 3, third box in Figure 5B). The system is thus switched to its active state and a new round of $D_{1}$ removal can start. Furthermore, *R* becomes sensitive to degradation. The series of two temporal switches separate the two timescales in the relaxation oscillations. The I/L switch sets the start of the slow timescale (the negative feedback loop), whereas the A switch marks the beginning of the fast clearance of $D_{1}$.

## 3. Discussion

### 3.1. A Novel Circadian Redox Oscillator Model

We have designed the first model for the complex biochemical system of Prx3-SO_2_H/Srx redox oscillations and we find that the loop Prx3-SOH – Prx3-SO_2_H, together with the negative feedback mediated by Srx (Figure 1B), is necessary and sufficient for the generation of oscillations. The design principles of this oscillator are (i) a fast Prx3-SOH inactivation followed by (ii) a slow and delayed negative feedback loop, where mitochondrial H_2_O_2_ leaks to the cytosol to promote the translocation of cytosolic Srx to mitochondria, approximately 9 h after the inactivation of Prx3-SOH. This simple backbone reproduces the previously described circadian oscillations of mitochondrial Prx3-SO_2_H and Srx as well as the delay in mitochondrial Srx import.

Although quantitative details of the kinetic parameters in the Prx3/Srx system remain largely unknown, we have found a reasonable set of parameters that is consistent with the biology they describe. Previous biochemical studies have estimated that Prxs are 1000–10,000 times more reactive to H_2_O_2_ than other reduced cellular thiols [18], and that thus they can confine the diffusion of the oxidant [23]. Under the assumption that 0.01% of the membrane area contains transport proteins (Aqp) that can passively transport H_2_O_2_ out of the compartment [22], the translocation rate of H_2_O_2_ is 10,000-fold smaller than its reduction rate (Table 1). This is in agreement with the parameter values, a=1000 and d=0.2. Since the import of Srx to mitochondria also requires a protein transporter, we have assumed this rate to be in the same order of magnitude as the translocation of H_2_O_2_ to the cytosol. The experimental determination of the rate at which a protein is transported through a membrane is challenging, because the transport depends on a number of factors including pH, membrane voltage, protein length, protein 3D conformation, membrane fatty acid composition, and ATP levels (if the transport is active), among others. Hence the estimation of translocation kinetics is only preliminary. Furthermore, the choices for Srx degradation rate *q* and hyperoxidized Prx3 reduction rate by Srx *b* are consistent with previous experimental studies, which have estimated a half-life of sulfiredoxin of 4–5 h [11,24], and a rate of reduction of hyperoxidized Prx by Srx of approximately 2 M^−1^s^−1^, 1000–10,000 times smaller than its oxidation rate [19] (Table 1).

It is known that relaxation oscillations typically depend on positive feedback loops [40,41]. However, even in the absence of explicit positive feedback loops, the redox oscillator model still produces oscillations of Prx3-SOH, Prx3-SO_2_H and mitochondrial H_2_O_2_ that are of relaxation type. Two temporal switches characterize their triangular-like waveform and divide the system’s dynamics into three phases. A summary is shown in Figure 6. This scheme with three phases and two switches is reminiscent to that of the cell cycle. A series of biochemical switches controls transitions between the various phases of the cell cycle. They maintain its orderly progression and act as checkpoints to ensure that each phase has been properly completed before progression to the next phase. Such switches have been shown to generate decisive, robust (and potentially irreversible) transitions and trigger stable oscillations [33,42,43]. It should be noted, however, that the switches that regulate cell cycle transitions are the consequence of a complicated network of positive and negative feedback loops and result in bistability, as opposed to our simple model that (strikingly) contains only one negative feedback loop. The redox model highlights the importance of negative feedbacks and the diversity of switches in the context of oscillations.

The detailed model (Appendix A) also oscillates along the lines of the design principles identified for the reduced model. It shows the same characteristics: relaxation oscillations, temporal switches, and expected period and I/R phase delay, supporting the robustness of the minimal core model. According to biochemical assays, the rate constant for Prx3-SH oxidation to sulfenic acid Prx3-SOH is in the order of 10^7^ M^−1^ s^−1^ [18]. From the simulations of the detailed model we see that when this rate ($k_{1}$ in the model) increases, the amplitude of Prx3-SO_2_H and Srx oscillations also increases (Appendix A). This suggests that although the loop Prx3-SH – Prx3-SOH – Prx3-SS – Prx3-SH is not required for oscillations, it fine-tunes the dynamics of the Prx3/Srx oscillatory system. Large Prx3-SO_2_H and Srx amplitudes are in agreement with experimental data. Western blots show intense bands at the peak of Prx3-SO_2_H or Srx oscillations and no bands at the trough [11].

### 3.2. Alternative Views on the Nature of the Negative Feedback in the Redox Oscillator

The oscillator model presented here applies to Prx oscillations in mammalian adrenal gland, heart, and brown adipose tissue. Prx daily rhythms have been found not only in mammals, but also in a variety of eukaryotes, cyanobacteria, and archaea, making these oscillation an evolutionarily conserved circadian rhythm marker [10]. A key unanswered question is what determines Prx oscillations in other cell types and phyla. Srx indeed accounts for the oscillations in adrenal gland, brown adipose tissue, and heart [11], but other organisms that display oscillations in Prx do not express Srx homologs (e.g., *C. elegans* and *N. crassa*), suggesting that other mechanisms might be in place. The same is true for mouse red blood cells. Although they express Srx, Prx-SO_2_H rhythms are largely unaltered in erythrocytes derived from Srx knockout mice and, by contrast, application of proteasome inhibitors eliminated the decay phase of the Prx-SO_2_H rhythm in wild-type mouse erithrocytes [12]. These findings suggest that proteasomal degradation might account for the decay phase and hence for the negative feedback of the Prx-SO_2_H oscillations in other cell types and phyla.

### 3.3. Crosstalk between Prx Rhythms and the TTFL in Eukaryotic Organisms

Given that Prx proteins and their oxidation rhythms are highly conserved among species, Prx rhythms seem to be evolutionarily ancient. It is also likely that they emerged before the circadian TTFL, which is believed to have evolved independently in separate species [10]. It is an important goal to now understand the relationship between the canonical TTFL and Prx oscillations, as it is becoming evident that there is a reciprocal interplay between the two. In one direction, the period length of Prx-SO_2_H rhythms is extended in embryonic fibroblasts derived from *Cry1/2*^−/−^ mice [8]. In behaviorally arhythmic *Drosophila* mutants and *N. crassa* mutants displaying a lengthened period, Prx oscillations are perturbed in phase [10]. In the other direction, *Arabidopsis* mutants lacking Prx genes display TTFL rhythms that are altered in phase or amplitude [10], and knockdown of specific PRX isoforms in human U2OS cells affects the period length and amplitude of clock gene rhythms [44].

The modeling approach described in this study assumes constant H_2_O_2_ production. Nevertheless, the sources of mitochondrial H_2_O_2_ in tissues where the redox clockwork is present have been shown to be, at least in part, under the control of the circadian TTFL. In adrenal gland, for example, the major source of H_2_O_2_ is steroidogenesis, and this process is regulated by the TTFL [45,46]. In heart or brown adipose tissue, in which oxidative metabolism is very high, the respiratory component becomes an important source of H_2_O_2_, and respiration rate has also been shown to be rhythmically controlled by the canonical TTFL [47].

Moreover, the Prx3/Srx redox oscillator model contains only one negative feedback, namely the reduction of Prx3-SO_2_H (*I*) by Srx (*R*), which is stimulated by the cytosolic H_2_O_2_ ($D_{2}$) increase. Cytosolic H_2_O_2_, however, can signal as a second messenger and activate the p38 MAPK pathway to decrease mitochondrial H_2_O_2_ production [11,17]. The periodic H_2_O_2_ release to the cytosol is expected to act on other cytosolic or nuclear targets that are not yet identified. Canonical clock proteins might be directly or indirectly affected by the cytosolic H_2_O_2_ increase. This idea is supported by a study from 2014, which showed that the interaction of the two mammalian core clock proteins PER2 and CRY1 is redox-sensitive [48], as well as by other studies that have shown that the cellular redox poise can regulate the TTFL oscillator [49,50]. The H_2_O_2_ leakage might thus represent a potential coupling node between both the redox and the TTFL clock. The reciprocal interplay between the Prx system and the local TTFL clock might allow synchronization between local metabolic activity and systemic circadian regulation. The presence of Srx as part of the Prx clockwork in adrenal gland, brown adipose tissue or heart, but not in the liver, for instance, might indicate differences in how metabolic signals are transmitted to the TTFL.

### 3.4. Model Predictions

Mathematical models are most useful when they make predictions that can be tested (and hopefully validated) in vivo. Because the Prx3/Srx oscillator model is not quantitative in detail due to the limited knowledge of rate constants, it only allows us to make some basic predictions. Interfering with the expression of Srx or specific aquoporin (Aqp) isoforms involved in the H_2_O_2_ translocation across mitochondrial membranes would, intuitively, disrupt Prx3/Srx oscillations. Moreover, the simulations show that H_2_O_2_ peaks in the cytoplasm ∼4.5 h after the peak of Prx3-SO_2_H (or ∼4 h after mitochondrial H_2_O_2_). This observation might represent an attractive target for redox sensors that could measure when H_2_O_2_ peaks in the cytosol in relation to its peak in mitochondria. The peak of cytoplasmatic H_2_O_2_ at a proper time might be of importance to transmit appropriately timed signals about the redox state of the cell to the canonical TTFL oscillator. It is likely that a stable phase relationship is required between both oscillators for their optimal function. Our model also predicts that the system should be able to oscillate even under conditions of constant mitochondrial H_2_O_2_ production. Such a self-sustained nature of the Prx3 cycle could be easily tested in isolated mitochondria led to respire in a constant environment.

It should be noted that the redox model reproduces the circadian rhythms at the level of mitochondrial membranes. Mitochondria vary in number depending on the cell type. For example, a human liver cell contains 1000–2000 mitochondria, whereas a cardiac myocyte contains 7000–10,000 mitochondria [51]. Although single mitochondria might be competent redox oscillators, it seems plausible that they synchronize their redox circadian cycles to each other, as seen by the clear bands in published western blots of Prx3-SO_2_H or Srx [11], in order to transmit the information about the redox state of the cell to the TTFL clock as an ensemble.

### 3.5. Concluding Remarks

Finally, we close with a short remark about the beauty of simple and generic models. We have elaborated on the basic ingredients that constitute the Prx3/Srx redox oscillator in mammalian mitochondria from adrenal gland, heart, and brown adipose tissue. We have found the core motif that is necessary and sufficient to generate self-sustained rhythms that reproduce previously published experimental data. Our model has been implemented using plausible and simple, yet generic mathematical, representations. This five-variable network might be relevant for other simplified oscillatory systems that follow the same logic of fast (in)activation followed by slow negative feedback, such as the mitotic oscillator involving cyclin and Cdc2 kinase interactions or the Ca^2+^ oscillations based on Ca^2+^-induced Ca^2+^ release [52]. The fact that nature has converged on common mathematical structures underlines the links between similar dynamic phenomena occurring in widely different biological settings. It even constitutes a step forward in the synthetic design of reliable biological clocks [53,54].

## 4. Materials and Methods

All temporal simulations and model analyses have been performed with Python, using the odeint integrator from the scipy module.

## Figures and Tables

**Figure 1 ijms-20-02368-f001:**
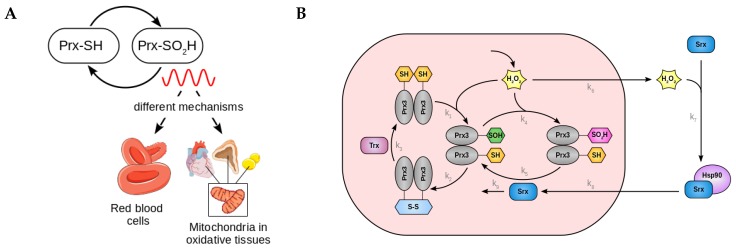
Peroxiredoxin (Prx) oxidation cycles occur with circadian periodicity in all kingdoms of life. (**A**) The mechanisms for 24 h rhythm generation in Prx hyperoxidation in mammals differ in different cell types. (**B**) Model for the mechanism underlying circadian oscillation of Prx3-SO_2_H and sulfiredoxin (Srx) levels in mitochondria of adrenal gland, heart, and brown adipose tissue. See text for details.

**Figure 2 ijms-20-02368-f002:**
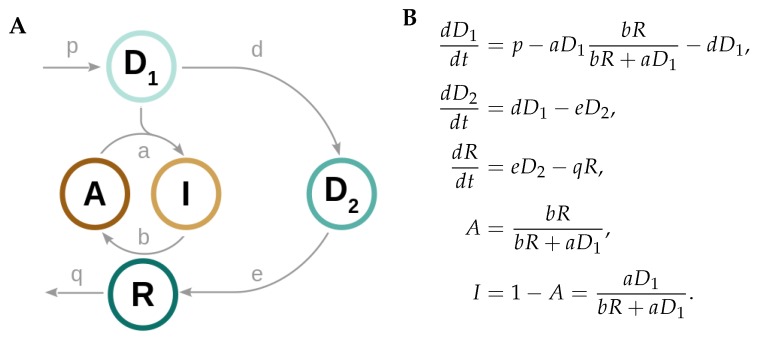
Core model for redox circadian oscillations. Scheme (**A**) and equations (**B**) of the core redox oscillator model. *A* represents active peroxiredoxin 3 (Prx3-SOH); *I*, inactive Prx3 (Prx3-SO_2_H); $D_{1}$, mitochondrial H_2_O_2_; $D_{2}$, cytosolic H_2_O_2_; and *R*, mitochondrial Srx.

**Figure 3 ijms-20-02368-f003:**
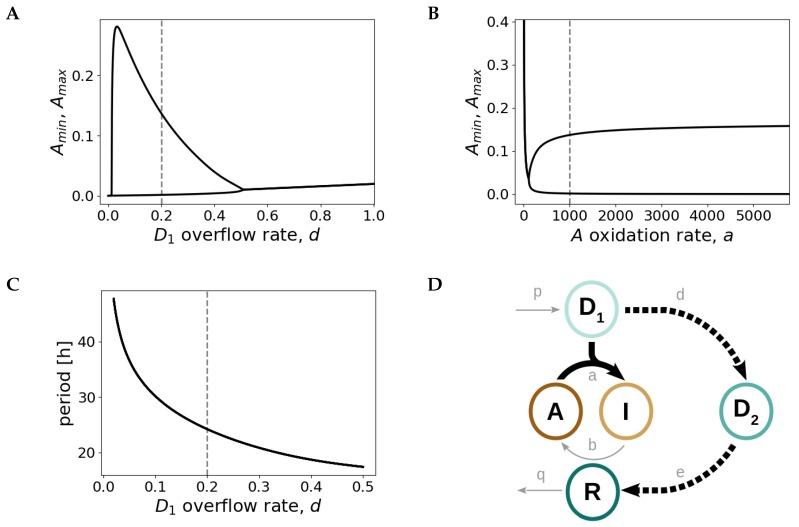
Fast *A* inactivation followed by a slow negative feedback loop is the design principle of the Prx3/Srx redox oscillator. Bifurcation diagrams as a function of the model parameters *d* (**A**) and *a* (**B**). At d=0.02 and a=120 self-sustained oscillations emerge. $A_{min}$ and $A_{max}$ represent the minimum and maximum values of *A* in the oscillatory regime. (**C**) Variation of period as a function of *d*. The curves shown in panels (**A**–**C**) are obtained for the default parameter set given in the caption of Figure 4. The dashed gray lines depict the default parameter values given in the caption of Figure 4. (**D**) Sketch of the core backbone for *I*/*R* redox oscillations: fast $D_{1}$-induced *A* inactivation (continuous thick line) followed by a slow $D_{1}-D_{2}-R$ negative feedback loop (dashed line).

**Figure 4 ijms-20-02368-f004:**
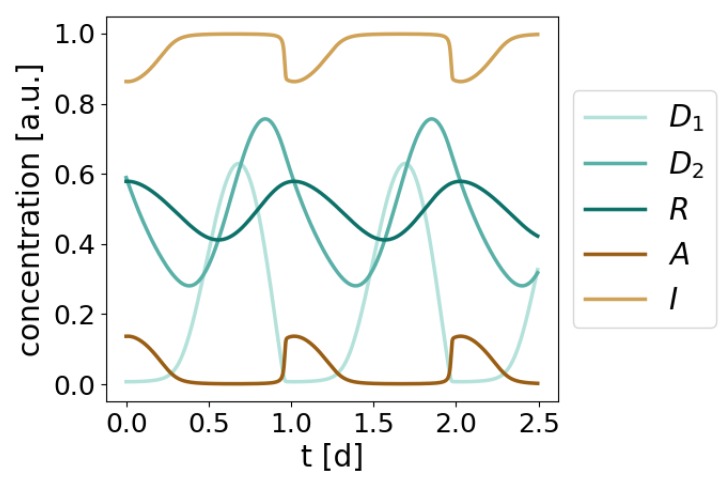
Dynamics of the redox model. Limit cycle oscillations obtained by numerical integration of the equations shown in Figure 2B for the following parameter values (arbitrary units, a.u.): p=1,a=1000,b=2,d=0.2,e=q=0.1. Consistent with experimental data [11] and with the model constraints (Table 1), the resulting period is 24.2 h and the phase shift between *I* and *R* is 8.7 h.

**Figure 5 ijms-20-02368-f005:**
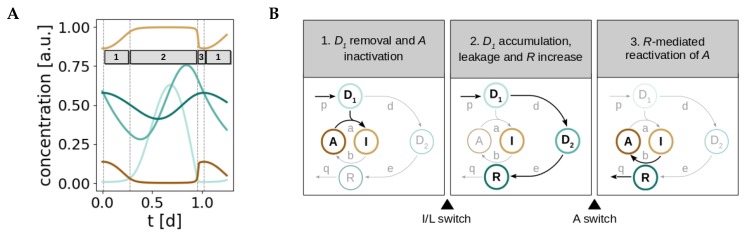
A series of three phases and two temporal switches can be regarded as the mechanism for generation of redox oscillations. (**A**) Division of the temporal dynamics of the I/R system into three phases that explain the major transitions of the model components. (**B**) Scheme of the major events that take place in each phase (see main text for details).

**Figure 6 ijms-20-02368-f006:**
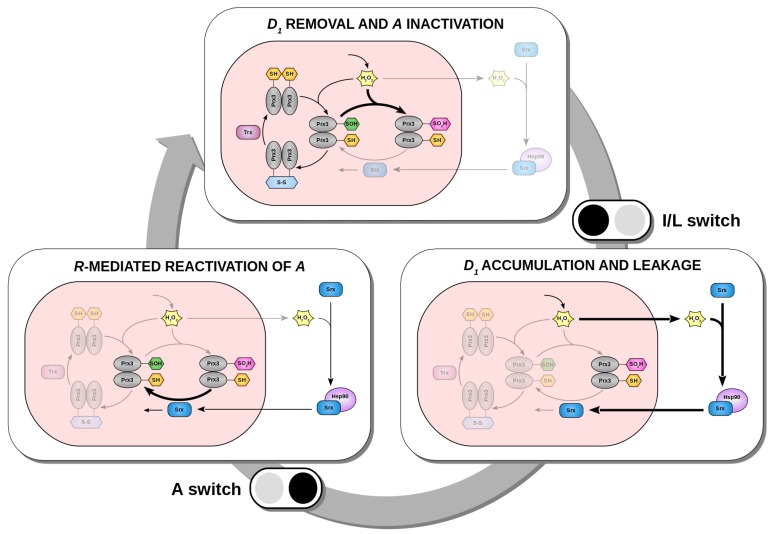
Model for the mechanism underlying circadian oscillations of Prx3-SO_2_H and Srx in mitochondria of mammalian adrenal gland, heart, and brown adipose tissue. Three phases and two temporal switches conceptually explain the generation of oscillations. The switches are represented by the black/gray icons. The switch occurs from black to gray: the black circle represents the phase from which the system comes, and the gray circle depicts the phase it progresses to.

**Table 1 ijms-20-02368-t001:** Physiological parameter constraints that allow finding plausible parameter ranges in the peroxiredoxin (Prx) 3/sulfiredoxin (Srx) oscillating system. The rates of Prx3 hyperoxidation (*I* formation), Srx-mediated Prx3 reduction (rescue of *A*), and Srx degradation (*R* removal) have been determined by enzymatic assays in previous studies. The rate of mitochondrial H_2_O_2_ translocation* is estimated from experimental and theoretical studies that have focused on H_2_O_2_ transport and signaling.

Model Parameter	Reaction Rate	Physiological Values	References
*p*	mitochondrial H_2_O_2_ production	-	-
*a*	Prx3-SOH hyperoxidation (and Prx3 inactivation)	10^4^ M^−1^ s^−1^	[18]
*b*	Prx3-SO_2_H reduction (and Prx3 reactivation)	2 M^−1^ s^−1^	[19]
*d*	H_2_O_2_ translocation to cytosol	1 s^−1^ *	[20,21,22,23]
*e*	Srx translocation to mitochondria	-	-
*q*	mitochondrial Srx degradation	0.16 h^−1^	[11,24]

* H_2_O_2_ transport through biological membranes is thought to occur via aquaporins (Aqp) [20]. The typical diameter of an aquaporin is ∼20 Å [21] and biological membranes typically contain 30 Aqp/µm^2^ [22] (although this number might vary across cell and membrane types). Thus, the total Aqp area that can transport H_2_O_2_ out of the mitochondria is ∼100 nm^2^ per µm^2^ of membrane. In other words, 0.01% of the membrane area contains Aqp. Very generally, the probability of an H_2_O_2_ molecule to diffuse can be assumed to be in the same order of magnitude of its probability to get reduced by Prxs [23]. With the limitation that only 0.01% of the membrane allows H_2_O_2_ translocation, the probability of H_2_O_2_ being transported out of the mitochondria is 0.01% of its probability of getting reduced, i.e., 10,000-fold lower than its reduction rate *a*.

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
