# Peer review of "A Robust Model for Circadian Redox Oscillations"

_ijms, 2019, doi:10.3390/ijms20092368_

Round 1
Reviewer 1 Report
GENERAL EVALUATION
This manuscript offers mathematical models for a provocative redox oscillator involving rhythmic changes in peroxiredoxin oxidation. This oscillation has been shown to be present in all phyla where it was tested (Edgar et al., Nature 2012, 485, 459–64). This is of interest because the standard transcriptional-translational feedback oscillators that are the focus of most of the field are not found in all phyla (eg, cyanobacteria and enucleated algae) and involve different protein components in different phyla. It was proposed that the peroxiredoxin oscillator may be the primordial circadian oscillator. In my view the principal argument against this possibility is that elimination of peroxiredoxin does not eliminate the TTFL oscillator (Edgar et al., Nature 2012, 485, 459–64); it merely alters the phase of the TTFL oscillation, suggesting that this oscillation is a mechanism for coupling redox inputs to the oscillator (what other outputs, if any, might it control?). In any event, the fact that it is possible to produce mathematical models for these oscillations that are persistent is of interest. The paper is well written and presents a detailed derivation of the simplified model that is the focus of the paper. The model assumes a rapid accumulation of the deactivated oxidized peroxiredoxin followed by a slow accumulation of cytosolic H2O2 that eventually triggers mitochondrial transport of Srx to reactivate peroxiredoxin; the combination of this rapid reaction with the slow reaction leads to oscillations. The paper will be of broad interest. I do have two points for the authors to address.
1) This oscillator model is apparently specific for some cells in mammals (adrenal gland, heart and brown adipose tissue), since other cells (eg red blood cells) do not show the dependence on sulfiredoxin (Srx) import to mitochondria. So how general will it be to peroxiredoxin oxidation oscillations across all phyla? What would replace the feedback by Srx for these other oscillators?
2) Mathematical models are most useful when they make predictions that can be tested (and hopefully validated) in vivo. Can the authors make any predictions from their model that could then be tested by others (in addition to the ranges for rate constants)? One possibility would be the nature of the Phase Response Curve for exogenously added Reactive Oxygen Species (ROS).
Author Response
RESPONSE TO REVIEWER #1
This manuscript offers mathematical models for a provocative redox oscillator involving rhythmic changes in peroxiredoxin oxidation. This oscillation has been shown to be present in all phyla where it was tested (Edgar et al., Nature 2012, 485, 459–64). This is of interest because the standard transcriptional-translational feedback oscillators that are the focus of most of the field are not found in all phyla (eg, cyanobacteria and enucleated algae) and involve different protein components in different phyla. It was proposed that the peroxiredoxin oscillator may be the primordial circadian oscillator.
Point 1: In my view the principal argument against this possibility is that elimination of peroxiredoxin does not eliminate the TTFL oscillator (Edgar et al., Nature 2012, 485, 459–64); it merely alters the phase of the TTFL oscillation, suggesting that this oscillation is a mechanism for coupling redox inputs to the oscillator (what other outputs, if any, might it control?).
Response 1: We thank the reviewer for this observation. As the reviewer has pointed out, given that Prx proteins and their redox rhythms are highly conserved among species (Edgar et al., 2012, Nature), Prx rhythms seem to be evolutionarily ancient. It is thus plausible that they appeared in life before the circadian TTFL, which has been proposed to evolve independently in different species. Mounting evidence is showing that there is a reciprocal interplay between canonical TTFL and Prx rhythms, and we have elaborated on these links in a new subsection inside the Discussion (section 3.3. “Crosstalk between Prx rhythms and the TTFL in eukaryotic organisms”, L291-322). We have also expanded on how cytosolic H2O2 (D2 in the model) represents an attractive node to couple the Prx clock the TTFL, which we plan to do in the near future.
In any event, the fact that it is possible to produce mathematical models for these oscillations that are persistent is of interest. The paper is well written and presents a detailed derivation of the simplified model that is the focus of the paper. The model assumes a rapid accumulation of the deactivated oxidized peroxiredoxin followed by a slow accumulation of cytosolic H2O2 that eventually triggers mitochondrial transport of Srx to reactivate peroxiredoxin; the combination of this rapid reaction with the slow reaction leads to oscillations. The paper will be of broad interest. I do have two points for the authors to address.
Point 2: This oscillator model is apparently specific for some cells in mammals (adrenal gland, heart and brown adipose tissue), since other cells (eg red blood cells) do not show the dependence on sulfiredoxin (Srx) import to mitochondria. So how general will it be to peroxiredoxin oxidation oscillations across all phyla? What would replace the feedback by Srx for these other oscillators?
Response 2: In the revised manuscript, we highlight from the beginning that this oscillator model is specific for some mammalian cells. We have modified two sentences in the abstract (L4-6) to point to the fact that our oscillator model applies to cells from adrenal gland, heart or brown adipose tissue, although Prx redox oscillations have been showed to occur in all kingdoms of life. We have also expanded on the nature of the negative feedback in other phyla as well as other mammalian cells apart from adrenal gland, heart and brown adipose tissue in the Introduction (L41-47) as well as in a new subsection in the Discussion (section 3.2. “Alternative views on the nature of the negative feedback in the redox oscillator”, L278-290). We thank the reviewer for pointing this important aspect that is now addressed in more detail.
Point 3: Mathematical models are most useful when they make predictions that can be tested (and hopefully validated) in vivo. Can the authors make any predictions from their model that could then be tested by others (in addition to the ranges for rate constants)? One possibility would be the nature of the Phase Response Curve for exogenously added Reactive Oxygen Species (ROS).
Response 3: We thank the reviewer for this point as well as for the nice sentence that we have included in the manuscript (L324-325). Following the her/his advice, we have added the subsection “3.4. Model predictions” in the Discussion (L323-346). Nevertheless, since the Prx3/Srx redox oscillator system is not quantitative enough due to limited rate constants, only some basic predictions are made, such as:
1) That the system should be able to oscillate under conditions of constant H2O2 production.
2) The model reproduces rhythms at the level of a single mitochondrion. But if indeed the system oscillates in the lab under conditions of constant H2O2 production, then synchronization among mitochondria is expected, since the western blot data published in (Kil et al., Mol Cell, 2015) shows very clear bands of Prx3-SO2H and mitochondrial Srx. Western blots are obviously a result of a bulk experiment (many cells and many mitochondria per cell) and, if the organelles were not synchronized, it would be very difficult to observe the clear rhythms in Prx3-SO2H or mitochondrial Srx.
3) That cytoplasmatic H2O2 should peak ~4h after mitochondrial H2O2 (if real time redox sensors could measure H2O2 on real time – so far there are none).
4) That genetic ablation of Srx or the Aqp isoforms involved in the translocation of H2O2 across mitochondrial membranes would result in the disruption of Prx3 oscillations.
We thank for the suggestion to study PRCs as a tool for model-experiment comparison. We plan to include PRCs in forthcoming studies.
Reviewer 2 Report
The paper of del Olmo and collaborators entitled « A Robust Model for Circadian Redox Oscillations » is proposing a deterministic model to explore circadian rhythm of peroxiredoxin oxidation in mitochondria of mammalian cells. The main conclusion is that time-lapse (or delay) is necessary for coherent circadian simulation and that mitochondria have to synchronize their activity within a single cell to allow proper functionning.
The main biological interest of this mathematical model is to open an hypothesis about synchronisation of mitochondria.
Among weak points, many metabolic data are not yet available. Consequently, authors are treating the complete known system and reducing their ambition to run a smaller model. Morevoer, the model derives clearly from seminal work of the same groupe on circadian clock (Pett JP et al, 2016 PLoS Comput Biol), explaining already the importance of delayed phase to sustain proper circadian rhythm.
My opinion is that authors have to be directed to a more mathematical journal because the connexion with molecular biology is weak. Synchronization between mitochondria is not really surprising for cell biologists and I don’t feel that the model can help foster new finding in the field.
Major modification before publication : the authors have to drastically improve the discussion to help readers from molecular biology to fully appreciate the interest of their model in circadian biology.
Author Response
RESPONSE TO REVIEWER #2
The paper of del Olmo and collaborators entitled « A Robust Model for Circadian Redox Oscillations » is proposing a deterministic model to explore circadian rhythm of peroxiredoxin oxidation in mitochondria of mammalian cells.
Point 1: The main conclusion is that time-lapse (or delay) is necessary for coherent circadian simulation and that mitochondria have to synchronize their activity within a single cell to allow proper functionning.
Response 1: We agree with the reviewer that, indeed, mitochondrial synchronization is essential for proper cellular function. Nevertheless, synchronization in mitochondrial redox state, as presented in the revised manuscript (L342-344), is a novel prediction in the context of circadian biology. Moreover, if it holds that single mitochondria are competent oscillators (as predicted by this model), then synchronization among mitochondria is expected, since otherwise, the Prx3-SO2H and mitochondrial Srx western blot data published by (Kil et al., Mol Cell, 2015) would be much noisier. However, the focus of our paper was to model the very basic mechanism that can lead to mitochondrial redox oscillations. Because redox synchronization is indeed an attractive conclusion and novel prediction, we have dedicated a paragraph to its explanation in the new subsection “Model predictions” in the Discussion (L338-346).
Point 2: The main biological interest of this mathematical model is to open an hypothesis about synchronisation of mitochondria.
Response 2: Indeed, our oscillator model can be extended to study synchronization of mitochondria. As discussed in the previous point, we have stressed this in the subsection dedicated to “Model predictions” in the Discussion (L338-346).
Point 3: Among weak points, many metabolic data are not yet available. Consequently, authors are treating the complete known system and reducing their ambition to run a smaller model.
Response 3: We agree that many metabolic data are not available, so we therefore reduced our extended model (Figure 1B or Appendix A) to a core mechanism (Figure 2A), that contains some known quantitative data. A summary of the availability of the experimental data is now included as Table 1, to improve readability.
Point 4: Morevoer, the model derives clearly from seminal work of the same groupe on circadian clock (Pett JP et al, 2016 PLoS Comput Biol), explaining already the importance of delayed phase to sustain proper circadian rhythm.
Response 4: There is indeed a similarity to the previously published delayed-differential equation model studied in (Pett et al., 2016, PloS Comput Biol) since in both systems delayed negative feedbacks drive self-sustained oscillations. However, the Pett model (which at the same time derives from previous work in our group (Korenčič et al., 2014, Sci Rep)) is based on transcriptional data of core clock genes, whereas our model of redox oscillations describes the (delayed) response to H2O2 and is motivated by protein data (Kil et al, 2015, Mol Cell). Moreover, the modeling approach is different: the Korenčič/Pett model makes use of the so called delayed-differential equations, whereas the model presented in this work is an ordinary differential equation model.
Point 5: My opinion is that authors have to be directed to a more mathematical journal because the connexion with molecular biology is weak. Synchronization between mitochondria is not really surprising for cell biologists and I don’t feel that the model can help foster new finding in the field.
Response 5: The reviewer points correctly to the problem that detailed quantitative data of the H2O2 dynamics are not available and thus this model is only a generic model. Nevertheless, in our eyes, a core model of redox oscillations with self-sustained rhythms is useful and can be of broad interest to circadian biologists. It serves as a proof of principle that the proposed mechanism (Kil et al., 2015, Mol Cell) leads to oscillations for reasonable parameters and can stimulate future experiments that we have addressed in the new subsection “Model predictions” from the Discussion (L323-346).
Point 6: Major modification before publication : the authors have to drastically improve the discussion to help readers from molecular biology to fully appreciate the interest of their model in circadian biology.
Response 5: We followed the advices of the reviewers and improved the discussion accordingly. The revised Discussion is now organized in different subsections that summarize the main findings of our work and that support the connection of the model with molecular circadian biology and physiology.
1) The first subsection (3.1. A novel circadian redox oscillator model L215-277) remains largely unchanged. It recapitulates the main findings of our model, i.e. that the combination of switches and negative feedback generates self-sustained oscillations that reproduce the oscillation characteristics observed experimentally (circadian period and a Prx3-Srx phase delay of ~8h). It also discusses how the parameter values, under which the system oscillates, are consistent with the reaction rates that have been measured via enzymatic assays. Our oscillator model can be regarded as a series of three phases and two switches that remind to the structure of the cell cycle and thus a short paragraph is dedicated to this analogy. Moreover, we expand on how the main ‘ingredients’ that make our minimal (5 variable) model oscillate also apply to the detailed model presented in Figure 1B and Appendix A.
2) Because the oscillator model presented in this work is specific for cells of mammalian adrenal gland, heart and brown adipose tissue, we have extended on what other mechanisms (and negative feedbacks) might be leading to Prx oscillations in other cell types and other kingdoms of life. This is presented now as subsection 3.2. in the revised Discussion (“Alternative views on the nature of the negative feedback in the redox oscillator”, L278-290).
3) The third subsection in the revised Discussion is dedicated to the crosstalk between Prx rhythms and the canonical TTFL, as it is becoming evident that there is a reciprocal interplay between both oscillators (L291-322, “Crosstalk between Prx rhythms and the TTFL in eukaryotic organisms”). The mathematical analysis of this interplay is work that we plan to do in the near future.
4) Because mathematical models are most useful when they make predictions that can be tested in the lab, we have included a subsection addressing the predictions that can be made from our generic redox oscillator model (L323-346, “Model predictions”).
5) We close the Discussion with a short remark about the utility of generic models, and how the two principles from the redox oscillator (fast inactivation and slow negative feedback that rescues the fast reaction) might also apply to other biological (natural or synthetic) oscillating systems.

Reviewer 3 Report
The manuscript titled “A Robust Model for Circadian Redox Oscillations” proposes the potential molecular mechanism for generating oscillations of Prx rhythms via computational modeling: the combination of switches and delayed negative feedback.
Major comments
1. L50-67: It is very difficult to understand the description of the system. Please add (i), (ii), (iii)…in both the Fig. 1B and text so that reader can recognize which part of Fig. 1B matches with the explanation in the text.
2. The key term of the model equation (Fig. 2B) is the nonlinear term abD_1R/(bR+aD_1). Please provide how this term is derived and underlying meaning.
3. L100-L117: Please provide a table which summarizes the parameter constrains for the readability.
Minor comments
1. L34: Peroxiredoxins => Prx
Author Response
RESPONSE TO REVIEWER #3
The manuscript titled “A Robust Model for Circadian Redox Oscillations” proposes the potential molecular mechanism for generating oscillations of Prx rhythms via computational modeling: the combination of switches and delayed negative feedback.
Major comments:
Point 1: L50-67: It is very difficult to understand the description of the system. Please add (i), (ii), (iii)…in both the Fig. 1B and text so that reader can recognize which part of Fig. 1B matches with the explanation in the text.
Response 1: Following the reviewer’s suggestion, we have adjusted Figure 1B as shown below (and in the revised manuscript) to name the reactions that occur in the Prx/Srx oscillating system (k1, k2, k3...). We have accordingly changed the text (L58-77) hoping that this will help the reader to be guided throughout the scheme. We also have included a reference to these reactions for example in L99 to facilitate understanding of the complex biochemistry.
Point 2: The key term of the model equation (Fig. 2B) is the nonlinear term abD_1R/(bR+aD_1).
Please provide how this term is derived and underlying meaning.
Response 2: We agree with the reviewer that derivation of the nonlinear term of the model and its meaning are not explained in the main text. We have included a sentence explaining that this nonlinearity arises from two bilinear terms via adiabatic elimination and that such mathematical derivation is further discussed in the Appendix B (L104-107). When discussing this nonlinearity in further sections, the reader is referred to Appendix B to understand the mathematics behind the derivation of the term (e. g. L107, L185).
Point 3: L100-L117: Please provide a table which summarizes the parameter constrains for the readability.
Response 3: We thank the reviewer for this constructive comment. We agree that a Table with the parameter values that were used to constrain the plausible ranges in the model is more easily read. We have included such Table in the revised manuscript (Table 1, see below) and modified the text accordingly, simplifying the paragraph where the constraints were listed (L120-124) and including references to the table in further sections (L157, L230, L240).
Minor comments:
Point 4: L34: Peroxiredoxins => Prx
Response 4: We have adjusted the language to alleviate having too many abbreviations at the beginning of the manuscript. Since ‘Prx’ is often used in the third paragraph of the manuscript (L34-47), we have introduced the abbreviation only at this point. We hope this will facilitate readability. We thank the reviewer for this observation.
The reviewer also suggested that Results and Discussion could be improved.
In the Results section, we have tried to guide the reader along the main findings in a clearer way:
1) By including names of the reactions in Figure 1B, as has been already discussed in Point 1.
2) By providing Table 1 with the parameter constraints, as discussed in Point 3.
3) By including references to which box of Figure 5B the main text refers to (L196-213).
4) We have also done some minor wording changes to shorten sentences (L107-113, L142-143), hoping that this will facilitate understanding.
Also, following the reviewer’s critique that the Discussion should be improved, we have have structured the discussion in different subsections and added some paragraphs to also reinforce the connection with circadian biology.

Round 2
Reviewer 2 Report
The manuscript of del Olmo et al has been significantly improved and deserves now publication without further modifications.
p { margin-bottom: 0.25cm; line-height: 115%; }
Author Response
RESPONSE TO REVIEWER #2
The manuscript of del Olmo et al has been significantly improved and deserves now publication
without further modifications.
We would like to thank the reviewer for his/her thoughtful comments and efforts towards improving
our manuscript to the level it is now.
p { margin-bottom: 0.25cm; line-height: 115%; }
Reviewer 3 Report
Indeed, the manuscript has been greatly improved. I have a couple minor points, which can authors can easily handles.
After going over response letter, I found why I was confused about the non-linear term in Fig. 2B. It should be written as "aD_1 \frac{bR}{bR+aD_1}". In this way, we can easily recognize that the non-nonlinear term is simply QSS of A.
Author Response
RESPONSE TO REVIEWER #3
Indeed, the manuscript has been greatly improved. I have a couple minor points, which can authors
can easily handles.
After going over response letter, I found why I was confused about the non-linear term in Fig. 2B. It
should be written as "aD_1 \frac{bR}{bR+aD_1}". In this way, we can easily recognize that the
non-nonlinear term is simply QSS of A.
We appreciate the interest that the reviewer has taken in our manuscript and the careful review and
constructive suggestions he/she has given. We fully agree that writing the nonlinear term in the
ODE for variable A as "aD_1 \frac{bR}{bR+aD_1}", as suggested by the reviewer, helps the reader
to understand that the removal of D_1 by means of A is modeled with mass action kinetics and
under the QSS assumption ofA. We thank the reviewer for this observation that we have revised in
the manuscript accordingly.
